# Play like me: Similarity in playfulness promotes social play

Jessica Frances Lampe[1], Sabrina Ruchti[1], Oliver Burman[2], Hanno Würbel [1],
Luca Melotti [1,3]*

**1** Division of Animal Welfare, University of Bern, Bern, Switzerland, **2** School of Life Sciences, University of
Lincoln, Joseph Banks Laboratories, Lincoln, United Kingdom, **3** Department of Behavioural Biology,
University of Münster, Münster, Germany

* luca.melotti@gmail.com

## Abstract

Social play is associated with the experience of positive emotions in higher vertebrates and
may be used as a measure of animal welfare. Altering motivation to play (e.g., through
short-term social isolation) can temporarily affect play levels between familiar individuals,
a process which may involve emotional contagion. This study investigated how forming
groups based on known differences in the personality trait "playfulness" (i.e., the longer-
term propensity of an individual to actively play from adolescence to early adulthood) affects
social play. Seventy-six adolescent male Lister Hooded rats underwent a Play-in-Pairs test
assessing their playfulness, ranked as high (H), intermediate (I) or low (L). At seven weeks
of age, rats were resorted into homogenous groups of similar (LLL, III, HHH), or heteroge-
neous groups of dissimilar (HII, LII) playfulness. Social play was scored in the home cage at
Weeks 8, 10, 12 of age. A second Play-in-Pairs test was performed (Week 11) to assess
consistency of playfulness. A Social Preference test investigated whether I rats in heteroge-
neous groups preferred proximity with I, H or L cage mates. It was found that heterogeneous
groups played less than homogeneous ones at adolescence (8 weeks of age), while play
levels at early adulthood (Weeks 10 and 12) did not differ between groups. Play in the homo-
geneous groups decreased with age as expected, while it did not change over time in the
heterogeneous groups, which did not compensate for the lower play levels shown at adoles-
cence. Play-in-Pairs scores before and after resorting were mildly correlated, indicating
some level of consistency over time despite the resorting procedure. In the Social Prefer-
ence test, subjects did not prefer one playfulness level over another. We conclude that a
mismatch in playfulness may negatively affect social play development, and thus the wel-
fare, of rats. Groups made of animals with similar playfulness, even those initially scoring
relatively low in this trait, seemed to be more successful in establishing play relationships
during adolescence.

pone.0224282

Pisa, ITALY

**Data Availability Statement:** All relevant data are
within the paper and its Supporting Information
files.

**Funding:** This study was funded by the Swiss
National Science Foundation (http://www.snf.ch/),

Project n. 31003A_144088 to HW and LM. The funder had no role in study design, data collection and analysis, decision to publish, or preparation of the manuscript.

**Competing interests:** The authors have declared that no competing interests exist.

## Introduction

Play behaviour occurs in most mammalian and several bird species [1, 2, 3]. Different types of animal play include locomotor, object and social play [4]. Social play, expressed in the form of rough-and-tumble play, has been most extensively studied in adolescent rats (e.g., [5, 6, 7, 8, 9]). During rough-and-tumble play, rats compete to access each other's nape, often resulting in one rat rotating into supine position to defend its nape and being pinned down by its playmate [8]. Such playful interactions are associated with the emission of specific, positively valenced, vocalisations [10, 11] and induce conditioned place preference in rats [12], suggesting a link between rough-and-tumble play and positive affective state.

More generally, animal play has been proposed as a tool to measure animal welfare [13, 14, 4]. During play, hedonic pleasure is experienced through the activation of the endogenous opioid system modulating the hedonic properties of rewards [15, 16]. As a result, play is autotelic [1] and can function as a reward like food and sex [17, 12, 18, 13, 10]. Though play often takes place in the absence of fitness threats [1], an increase in play may also occur as a way of coping with specific contexts such as, for example, competitive pre-feeding situations [19], social tensions with unfamiliar individuals [20], or a reduction in maternal care [21]. Taken together, while the association between play and animal welfare is context-dependent, there is converging evidence that play is associated with the experience of positively valenced emotional states *via* motivational and hedonic mental processes [4, 15].

As social play involves continuous interactions and (mainly) close physical contact between players, the question arises whether positive emotional states experienced by an individual can be conveyed to the rest of its social group through play. The process of emotional contagion can be regarded as a basic level of empathy through which the emotional state of an observer changes as a direct consequence of perceiving and sharing the emotional state of another [22, 23, 24]. Certain forms of emotional contagion can be mediated by rapid mimicry, one of the phenomena at the basis of the synchronisation of behavioural and emotional responses [23]. Rapid mimicry is a fast and involuntary response involving the cortical mirror neuron system, which processes both the appraisal and the response to emotion-related stimuli [25]. By automatically mimicking an action or facial expression of another individual, the observer may rapidly experience the emotional state related to that behaviour [22].

The link between emotional contagion and animal welfare is straightforward. If emotional states can be transferred rapidly and repeatedly between individuals, the emotional responses (either negative or positive) of even just a few individuals could affect the emotional state, and thus the welfare, of the entire social group [26, 4]. That is, not only what an individual feels, but also to what extent group members are influenced by its pleasure or distress is relevant for animal welfare [27]. For example, the synchronisation of behavioural responses and the sharing of emotional states have been suggested to promote the building and consolidation of social bonds [26, 28] which can lead to long-term welfare benefits. Furthermore, the frequency of rapid mimicry has been found to increase with the level of familiarity between subjects. For instance, humans and chimpanzees are more likely to mimic the facial expressions of in-group- compared to out-group members (reviewed in [29]). Within group rapid mimicry is also affected by the quality of the relation between interacting subjects (e.g., human friends mimic each other's smiles more than non-friends [30]). In this light, the replication and the sharing of fellows' emotions promote the development of tighter social relations, which contributes to social cohesion, one of the factors promoting animal welfare. The tendency of social animals to match emotions has been mainly demonstrated in situations linked to negative emotions (Fear: [31, 32]; Restraint distress: [33]; Depression: [34]; Pain: [35]), yet positive emotional contagion should be taking place as

well [36], as some studies have demonstrated in humans (e.g., [37, 38, 39]), primates [40, 41] and dogs [42, 43].

In rats, only a few studies have provided evidence of emotional contagion through play behaviour. Based on observations of social play between pairs of rats, Pellis & McKenna [7] found that a highly playful rat induced its partner to play more, and conversely a non-playful rat led its partner to play less, suggestive of contagion. Similar results were obtained by pairing group-housed rats having lower motivation to play with isolation-housed or temporarily isolated rats having higher motivation to play [9, 44]. The same studies, however, also showed that when there was a mismatch in play motivation, the less playful rat avoided its partner more often if given the chance to do so [9, 44]. In view of the conflicting evidence from the literature, further investigation is needed to understand how play (dis)similarities may affect the welfare of individuals within the same group. Moreover, in the above studies emotional contagion was assessed under *ad hoc* test conditions rather than in the home cage (conditioned place preference tests: [9]; play in pairs tests: [44, 7]) and individual differences in motivation to play were experimentally induced through social isolation (1 to 21 days of isolation before testing: [9, 44, 7]) rather than measuring differences in undisturbed, spontaneous social play within the home environment. Assessing the development of spontaneous social play across a relatively long period of time (i.e., from adolescence to early adulthood in this study) could better explain the link between play behaviour and animal welfare. Indeed, play can be regarded as a tool to assess and promote animal welfare only if its contagious properties are expressed consistently across at least part of a lifetime. That is, only if play is investigated as the personality trait "playfulness" can we determine long-lasting positive (or negative) effects of play on animal welfare. Playfulness in a social context can be defined as the intrinsic propensity of an individual to actively play with other group members [45].

Very few studies have looked at playfulness in rats. Pellis & McKenna [7] found that pairs of rats, and to a lesser extent the individuals within the pairs, showed consistent play levels across adolescence. Similarly, individual social play levels measured in a "Play-in-Pairs" test were found to be consistent between early and late adolescence and across motivational contexts (i.e., with or without prior social isolation; [45]). So far no study has investigated emotional contagion within social groups that were formed based on prior assessment of the playfulness level of each individual. This approach allows for the systematic exploration of play development over time based on known initial differences in playfulness between group members.

Using the Play-in-Pairs test [46, 45], we ranked groups of rats according to playfulness (H = high, I = intermediate, L = low) and resorted them to form five treatment groups. Three had similar playfulness within group yet varying levels of playfulness between groups ("homogenous" groups: HHH, III, LLL), and two had dissimilar playfulness within group, with either an H or L seeder rat housed together with two I rats ("heterogenous" groups: HII, LII). In the heterogeneous groups, I rats were used as a means of testing for both increases and decreases in play depending on the playfulness of the heterogeneous partner. We then assessed home cage play levels between adolescence and early adulthood and tested whether these were affected by the treatment groups.

Based on the available literature, we hypothesised that (i) the HHH group would play the most, being formed by highly and similarly playful individuals [9, 7, 47] and that (ii) the III group would play more than the LLL group. With respect to the heterogeneous groups, we expected that (iii) the HII group would play more than the LII group, as the H seeders would be expected to increase the play levels of their cage mates, while the L seeders to decrease them [44, 7]. To better assess whether the I rats preferred to spend time together with their seeder (H or L) or with their I cage mate, a social preference test [48] was also performed. Finally, we

hypothesised that (iv) homogenous groups (at least those with high or intermediate playfulness) would play more than the heterogeneous groups, as cage mates with similar motivation to play have been found to avoid each other less [9, 44].

## Methods

### Ethical statement

This study was conducted in compliance with the Swiss regulations on animal experimentation and formally approved by the Veterinary Office of the Canton of Bern (License no. BE 17/13).

### Animals and housing

The subjects were 76 male Lister Hooded rats born from 25 litters at Charles River Laboratories, Sulzfeld, Germany, and transported at weaning (21 ± 1 days) to the Division of Animal Welfare, University of Bern, Switzerland. Male rats were chosen because they exhibit higher levels of social play than females during adolescence [8, 9]. Upon arrival, the 76 rats were assigned to 19 cages of four non-littermates so that all cages had different litter combinations. Moreover, when picking up rats from their litter compartments, the pick-up order was counterbalanced across cages to avoid bolder and/or shyer rats to end up in the same cage. This was achieved by sorting into different cages those animals that escaped less from the approaching hand of the experimenter. Animals were housed in "Mickey 2 XL" cages (l × d × h: 80 × 50 × 38 cm; Savic, Belgium) distributed across two housing rooms. Two hammocks hanging from the top of the cage and a running wheel were provided to each cage. Cage bedding consisted of wood chips (JRS Lignocel), with three paper towels and three wooden tongue depressors (for shredding/chewing) provided weekly as enrichment. Animals had *ad libitum* access to standard rodent food (KLIBA NAFAG, Switzerland) and tap water. Housing room temperature was maintained between 21–24˚C and humidity between 50–70%, with a reverse 12:12 light: dark cycle (lights off at 9 am) allowing us to carry out behavioural tests and observations during the dark, active phase for the rats. Initial habituation to being handled by the experimenter was provided during the three days following arrival to the lab (10 minutes a day per cage). As part of a parallel study investigating positive facial expressions in the same rats, all rats were extensively habituated to being handled, tickled and photographed by the experimenters throughout the study period [49]. These procedures involved gradual and short (a few minutes per day) exposures to white light during the dark phase in order to take photos of the face, and were not performed on the days when the tests and observations of the present study occurred. This was done to ensure that such procedures did not interfere with the conspecific play measures taken in this study.

All animals were tested during the dark phase and were identified by their pelage patterns.

### Experimental design overview

During the fifth week of age, a conspecific "Play-in-Pairs" test was performed to rank the rats according to their playfulness (H = high, I = intermediate, L = low; see below section for details). After two weeks (seventh week of age), rats were resorted into new groups of three unfamiliar non-littermates based on their playfulness level. Five treatment groups of five cages each were formed (25 cages in total). Three homogeneous treatment groups consisted of cages with similar playfulness (intermediate playfulness: III; low playfulness: LLL; high playfulness: HHH), while two heterogeneous treatment groups consisted of cages with dissimilar playfulness (HII and LII). The H or L rats from the heterogeneous groups were named 'seeders'.

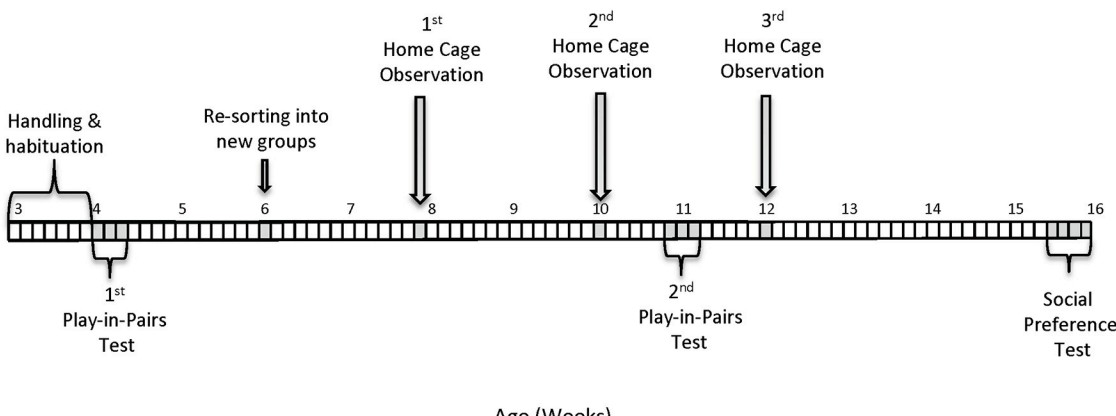

**Fig 1. Experimental timeline.**

Following the resorting based on playfulness, home cage play was assessed at three time points, namely at 8, 10 and 12 weeks of age. At the eleventh week of age, a second Play-in-Pairs test was performed to assess the consistency of playfulness after the resorting of the animals into new groups, and to make a comparison with the play levels observed in the home cage. Lastly, the I rats from the dissimilar treatment groups performed a social preference test for assessing their preference to spend time in proximity of either the seeder or the I cage mate. The experimental timeline and a summary of each experimental phase are presented in Fig 1 and S1 Table.

## Play-in-Pairs tests and resorting based on playfulness

A first Play-in-Pairs test was performed during the fifth week of age and across three consecutive days to assess individual playfulness levels [46, 45]. It consisted of repeated play sessions using all possible pair combinations of cage mates available (i.e., six pair combinations per cage, as the cages were comprised of four rats), which resulted in three play sessions per rat that were used to determine individual playfulness. This procedure provides a measure of individual playfulness that is at least partly independent from the playfulness of the play partners [7, 46, 45].

Habituation to the test arena (standard transparent Type IV cage, 59.5 × 38.0 × 20 cm; Tecniplast, Italy) and to a transport cage (standard transparent Type II cage, 26.7 × 20.7 × 14 cm; Tecniplast, Italy) began two days prior to testing. Rats were individually placed in the transport cage and taken to the test arena in which they stayed for 5 and 7 minutes on the first and second day, respectively. On each test day, all rats underwent a 3.5 h period of social isolation immediately prior to the Play-in-Pairs test to increase their motivation to play [50]. The isolation time of 3.5 h was chosen as it has been shown to induce a half-maximal increase of play in rats [51]. Animals were socially isolated in the same Type II cages used for transportation (see details above), with wood shavings for bedding, a paper towel, a few chocolate sprinkles (Deruijter, Netherlands), *ad libitum* standard food pellets (KLIBA NAFAG, Switzerland) and tap water provided. Each rat was tested once a day, and was paired with a different cage mate on every testing day. A randomised and counterbalanced trial order was applied, so that the six pair combinations of each cage were evenly balanced across time of day. In each testing trial, a pair of cage mates was placed in the arena for 10 minutes, during which frequencies of attacks to the nape (AN) and pinning (PN) behaviour were scored for each rat. AN as a measure of play initiation, and PN as the most frequent resulting play outcome, have been regarded as the

most representative measures of social play in rats [52, 5, 8]. AN was scored when a rat launched in the direction of another rat's nape and had its snout and / or front paws touching or very close to contact with the nape of the other rat. A PN event was scored when a rat pounced on its cage mate while this was completely rotated onto its back, and landed with both front paws on the supine rat's chest or stomach [45]; every instance of the attacker raising its paws and pouncing again onto the already rotated rat was counted as a new event. This definition of PN aimed at obtaining a better measure of how motivated to play (or positively aroused) the attacker was. In order to assess the level of symmetry between playful pins performed and playful pins received by a rat (according to the above definition of PN), a Play-Role index was calculated using the following formula: $\left|\frac{p-r}{p+r}\right|$, where $p$ is the frequency of pins performed and $r$ is the frequency of pins received for each pair of play partners [46]. A higher Play-Role index indicates that pins are mainly performed by one of the two play partners (suggestive of a stronger hierarchy between them), while a lower index indicates a greater reciprocation of playful pins between them, which is typical of juvenile play [8]. Trials were video-recorded using infrared cameras (VC Videocomponents GmbH, Germany) supplemented by infrared lamps. Videos were analysed blind to treatment group and to playfulness ranking of individuals using Interact software (Version 14, Mangold, Germany).

To determine the playfulness level of each rat, its social play frequencies (AN + PN) on the three consecutive testing days were summed to form individual playfulness scores, which were then ranked from highest to lowest playful animals. Rats were divided into groups corresponding to low- (L), intermediate- (I), and high- (H) playfulness categories (S2 Table). At six weeks of age, rats were resorted into cages of three unfamiliar, non—littermate rats and assigned to five treatment groups (LLL, III, HHH, HII or LII), each group consisting of five cages. Treatment groups were counterbalanced across the two housing rooms and shelf levels. The seeders selected for the heterogeneous HII and LII groups were the five rats with the highest playfulness scores and the five rats with the lowest playfulness scores, respectively. These rats were grouped with I rats from the middle of the I category (S2 Table). For the homogeneous HHH, III and LLL groups, the remaining rats from each playfulness category were sorted so that within each cage rats had the closest possible playfulness score.

One rat was excluded from the study at this point to achieve even distribution in the cages (i.e., 25 cages of three rats).

At the 11th week of age, after a day of re-habituation to the play arena (6 min per rat), a second Play-in-Pairs test was repeated among the resorted cage mates to assess the consistency of playfulness across time, and for comparison with home cage play levels.

## Home cage play observations

Continuous observation of play behaviour in the home cage was conducted at Weeks 8, 10 and 12 of age during the second quarter of each of the first four hours of the dark phase (10.15–10.30 h, 11.15–11.30 h, 12.15–12.30 h and 13.15–13.30 h), resulting in 1 h observation per animal and age point (225 h of focal observation in total). This sampling procedure was chosen based on continuous pilot observations of a subsample of cages, which revealed the presence of play peaks of relatively short duration, which were interspersed across the first four hours of the dark phase. Also, it was previously shown that a 1 h observation per day during a highly active period can allow for the collection of relatively high frequencies of play [46]. For this reason, observations focused on the first hours of the dark phase as during this period general activity levels are relatively high [53]. Home cages were video-recorded using infrared cameras (VC Videocomponents GmbH, Germany) supplemented by infrared lamps. Videos were analysed blind to treatment group and to playfulness ranking of individuals using Interact

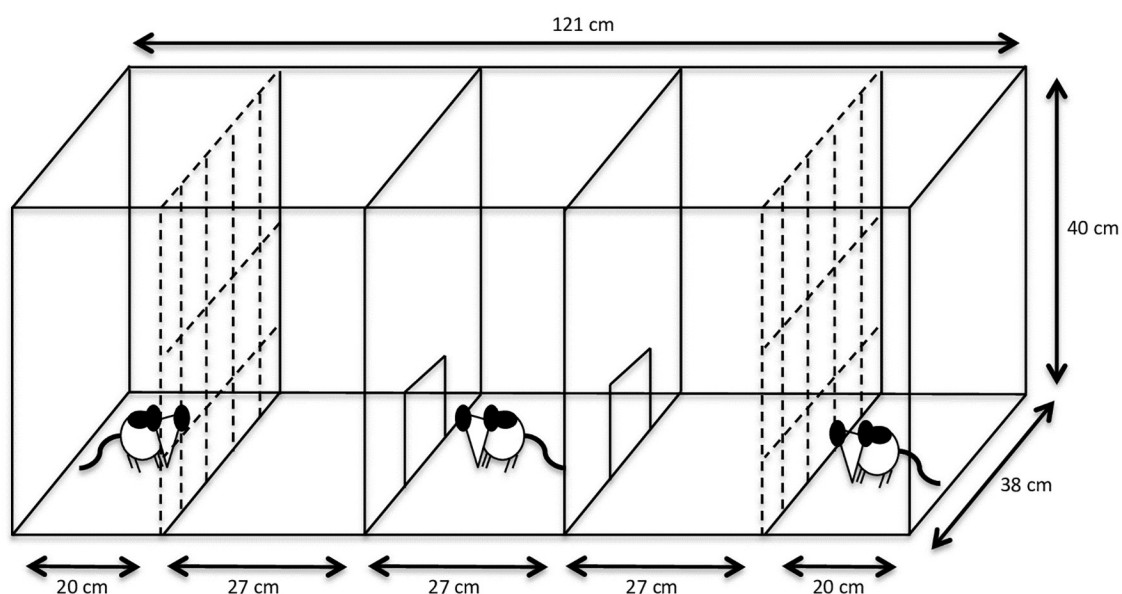

**Fig 2. Social preference test apparatus.** The subject rat was placed in the middle chamber and could choose to enter two adjacent chambers in which a stimulus rat, separated from the subject rat by a grid, could be approached. This test assessed whether the subject rat preferred to spend time in proximity with a rat of similar (I) or dissimilar (H or L) playfulness.

software (Version 14, Mangold, Germany). The frequencies of AN and PN were recorded for each rat and the Play-Role index was calculated in the same way as for the Play-in-Pairs tests (see section above).

## Social preference test

A social preference test [48] was performed by the I rats of the heterogeneous treatment groups (N = 20) in order to investigate whether these rats preferred to spend time in proximity of a "stimulus" cage mate with either similar (I) or dissimilar (H or L) playfulness. It was hypothesised that partners of higher playfulness would be preferred. The arena consisted of a modified social preference test apparatus [54] made of solid plastic and comprising five adjacent chambers (Fig 2). Subject animals (I rats) were individually placed in the middle chamber and had access to the two adjacent chambers on each side through a 10 × 10 cm opening. Stimulus rats were placed in the two outer chambers, which were separated from the adjacent chambers by a grid that allowed for olfactory and limited physical (i.e. sniffing) contact between subject and stimulus rats. Clear acrylic glass sheets were placed above the arena to keep the rats from climbing out the test apparatus. Habituation took place across two days. On the first day, in the morning, all cage mates together were habituated to the middle parts of the arena for 5 minutes. In the afternoon, I rats spent 2 minutes individually in each of the three middle chambers (6 min in total) and the seeders spent 2 minutes individually in the two outer chambers (4 min in total). On the second day, each rat spent 7 minutes in each of its possible test situations.

The test spanned across 4 days and each subject rat performed two 10-min trials (either on Days 1 and 3, or on Days 2 and 4). For the HII treatment group, trial partners were the H and I cage mates while, for the LII treatment group, trial partners were the L and I cage mates. To control for possible left- or right-side biases, the side of the stimulus seeder rat was counterbalanced within each cage. That is, when an I subject rat had the seeder rat on the right side in its first trial, the other I subject rat from the same cage had the seeder rat on the left side in its first

trial. In the second trial, the sides were reversed. On every testing day, cages were tested in a randomised order. The time (s) spent by the subject rat in each of the chambers adjacent to a stimulus rat was recorded across the two test trials. The difference between the time durations in these two chambers, indicating a preference for either the seeder or the I stimulus cage mate, was used for the analysis. A subject rat was considered to have entered a chamber when all four paws of the rat had crossed the opening connecting two chambers. Time durations were recorded live and blind to the side location of the seeder rat.

## Data analysis

Statistical analyses were performed using SPSS (SPSS®, version 22). A multilevel mixed-effects linear model was used, with PN as the dependent variable, cage as a random factor, treatment group and the interaction between treatment group and time as fixed factors, and time as a covariate. The maximum likelihood method (ML) was applied to build the model, using -2LL as goodness-of-fit index. Because a cross-level interaction was examined, group mean centring was used [55]. Post hoc, the estimates of fixed effects and the estimates of covariance parameters were calculated. An analogous mixed-effects linear model was conducted replacing the 5-level factor "treatment group" with the 2-level factor "group type" (homogeneous vs. heterogeneous treatment group types). When the covariance parameter estimate of cage showed that this factor was not a significant predictor in the model (as indicated by the Wald $z$ test not being significant), the cage factor was removed from further post-hoc comparisons. Based on the measured effects of the fixed factors and their interaction, and on visual inspection of the data, appropriate comparisons (independent / paired t-tests or Mann-Whitney U / Wilcoxon tests with Bonferroni adjustment for multiple comparisons, and an ANOVA with post-hoc comparisons using Sidak adjustment) were made to assess differences between and within time points. The dependent variable AN was not analysed as it positively correlated with PN (see Results).

To compare play levels of seeders with those of I rats in the heterogeneous groups, Mann-Whitney U tests and Friedman tests were used. The play frequencies of the pairs of I cage mates were averaged in order to have a balanced comparison between seeders and I rat pairs.

Intra-observer reliability of the play measures recorded in the Play-in-Pairs test after resorting, and in the home cage during the 15-min observation periods, was assessed by rescoring one randomly chosen video for every set of five videos (20% of the videos). Bland-Altman plots using 95% limits of agreement (i.e., mean difference of the two measurements ± 1.96 standard deviation of the mean difference) were used together with linear regressions using the difference of two measurements as dependent and the mean of both measurements as independent variable. A significant $t$-score of the regression would indicate a proportional bias in the Bland-Altman plot (i.e., a trend towards more scores below or above the mean difference line). In addition, one-sample t-tests, using the difference of the two measurements as variable, were performed to check whether the two measurements significantly differed from 0.

The analysis of the Play Role indexes required the use of cage as statistical unit, since cage composition changed when the animals were resorted according to playfulness, therefore the Play Role indexes of cage mates were averaged. The indexes of the first and second Play-in-Pairs tests were compared using a Mann-Whitney U test, while paired-tests were used for comparisons of measures taken after the resorting of the animals. Namely, a Friedman´s test assessed the change of Play Role index across the three time points of home Cage play, whereas a Wilcoxon Signed Rank test was used to compare the indexes of the second Play-in-Pair test with those of home Cage play (pinning frequencies of all time points summed together).

For the social preference test involving the heterogeneous groups (HII and LII), One-Sample Wilcoxon Signed Rank tests were run to examine whether the median of the differences between the durations spent in the two chambers adjacent to the stimulus rats (time spent adjacent to seeder—time spent adjacent to I rat) differed from 0, which would indicate a preference for either the seeder or the I stimulus rats. Kruskal-Wallis tests were run to check for significant cage effects.

The relationships between play measures were assessed using partial correlations controlling for the nested factor cage. Spearman's correlation was used if data did not meet parametric assumptions even after transformations; in this case, an effect of cage on the correlated variables was checked with a Kruskal-Wallis test.

## Results

### Intra-observer reliability of Play-in-Pairs test and home cage measures

For the Play-in-Pairs test after resorting, the Bland-Altman plot and tests indicated good intra-observer reliability for both AN ($t_{29}$ = -0.40, p = 0.70; mean differences ± SD: -0.03 ± 1.22; one-sample $t_{29}$ = -0.15, p = 0.88) and PN ($t_{29}$ = -1.42, p = 0.16; mean differences ± SD: 0.06 ± 1.88; one-sample $t_{29}$ = 0.20, p = 0.84). Similarly, the home cage play measures AN ($t_{113}$ = 0.18, p = 0.86; mean differences ± SD: 0.01 ± 0.70; one-sample $t_{83}$ = 0.14, p = 0.88) and PN ($t_{113}$ = 1.18, p = 0.24; mean differences ± SD: -0.02 ± 0.54; one-sample $t_{83}$ = 0.00, p = 1.00) also had good intra-observer reliability.

### Home cage play levels across treatment groups and time

AN and PN were highly correlated ($r_{72}$ = 0.68, p < 0.001). Therefore, only PN was used for further analyses, being the most frequent play measure (mean ± SD; PN: 39.27 ± 26.33; AN: 30.53 ± 18.94).

Time and treatment group significantly interacted ($F_{4,150}$ = 4.62, p < 0.01; Fig 3). The main effects of treatment group ($F_{4,205.48}$ = 3.76, p < 0.01) and time ($F_{1,150}$ = 23.84, p < 0.001) were also significant. Post-hoc comparisons based on the estimates of fixed effects showed that the development of play over time (slopes across all three time points) differed between HII and each of the homogeneous groups (HII vs. HHH: $t_{150}$ = -2.20, p = 0.02; HII vs. III: $t_{150}$ = -2.18, p = 0.04; HII vs. LLL: $t_{150}$ = -3.84, p = <0.001). They also differed between LII and LLL ($t_{150}$ = -3.28, p = 0.001), yet not between LII and III ($t_{150}$ = -1.60, p = 0.12), or LII and HHH ($t_{150}$ = -1.64, p = 0.10) groups. No differences were found among homogeneous groups (III vs. LLL: $t_{150}$ = -1.68, p = 0.10; III vs. HHH: $t_{150}$ = -0.04, p = 0.98; HHH vs. LLL: $t_{150}$ = -1.64, p = 0.10) or between heterogeneous groups (HII vs. LII: $t_{150}$ = -0.57, p = 0.57).

According to the estimates of covariance parameters, there was no significant cage variation (Wald $z$ = 1.52, p = 0.12), thus the random cage factor could be disregarded in further comparisons. From visual inspection of the data, the interaction between time and treatment group seemed to be mainly explained by differences between treatment groups at Time 1. Therefore, time was treated as categorical variable and multiple comparisons were performed to assess differences between and within time points. Homogeneous groups decreased or tended to decrease play over time (Time 1 vs. Time 3: LLL, $t_{14}$ = 4.82, p < 0.001; III, $t_{14}$ = 2.70, p = 0.02; HHH, $z$ = -2.13, p = 0.04; Time 1 vs. 2: LLL, $t_{14}$ = 2.78, p = 0.015; for the other comparisons of Time 1 vs. 2 and of Time 2 vs. 3, p ≥ 0.26; Bonferroni adjustment: α = 0.017). By contrast, play levels in the heterogeneous groups did not change across time (for all comparisons of Time 1 vs. 3, 1 vs. 2 and 2 vs. 3, p ≥ 0.10). In addition, at Time 1 the LLL group played more than the HII group ($t_{20.42}$ = -4.40, p < 0.001) and the LII group (U = 38.5, z = -3.08, p = 0.001). For all

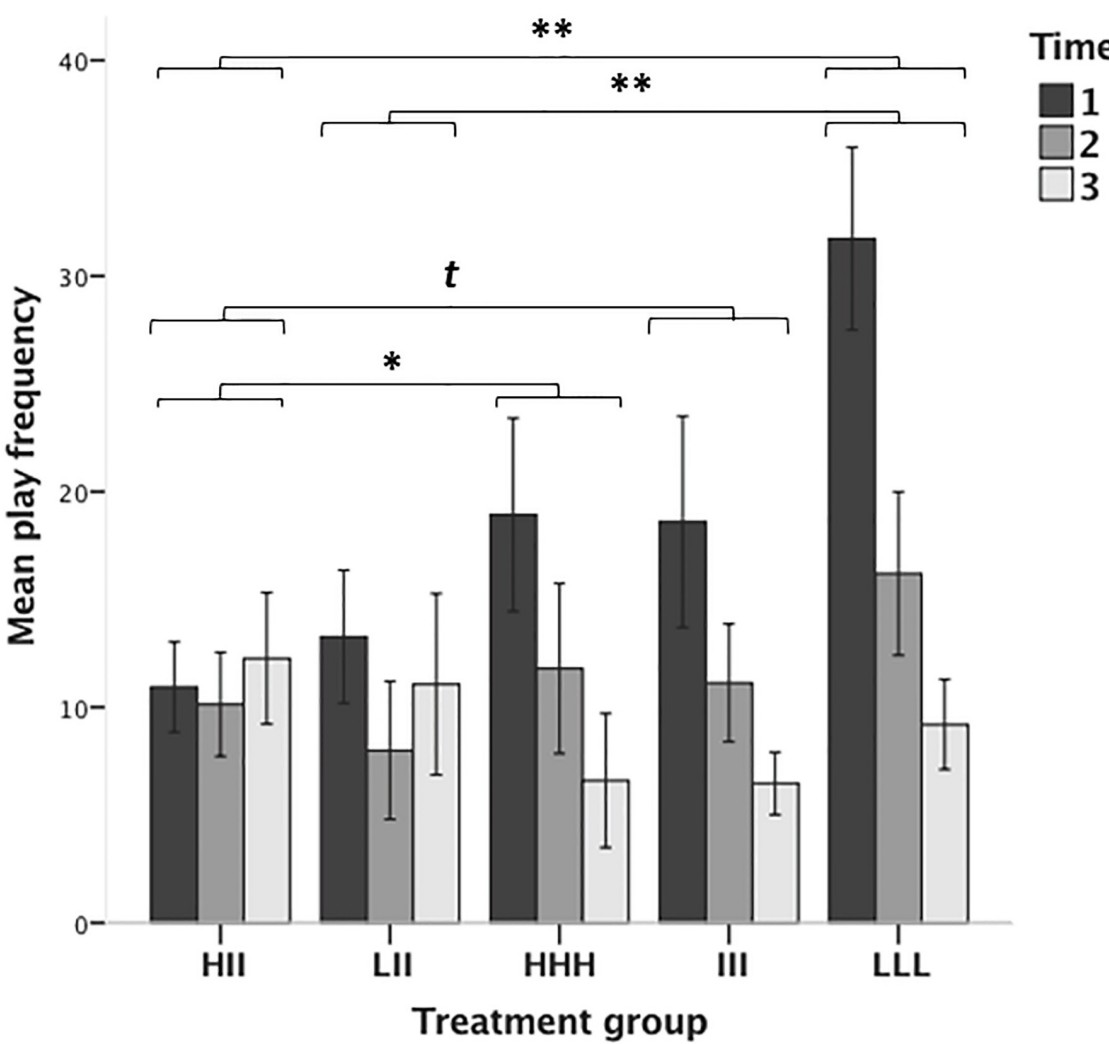

**Fig 3. Histograms showing home cage play levels (pinning frequency) across treatment groups and time; given are means and S.E.M.s.** Brackets indicate differences between treatment groups ($t$: $p < 0.1$; $^*$: $p < 0.05$; $^{**}$: $p < 0.01$).

other group comparisons at Time 1, $p \geq 0.02$ (Bonferroni adjustment: $\alpha = 0.005$). The treatment groups also did not differ at Time 2 or at Time 3 (all $p \geq 0.10$).

With regard to the main effect of treatment group and based on the estimates of fixed effects (i.e., with all time points taken together), the HII group played less than the HHH ($t_{205.48} = 1.98$, $p = 0.04$) and LLL ($t_{205.48} = 3.46$, $p = 0.001$) groups, and also tended to play less than the III group ($t_{205.48} = 1.96$, $p = 0.06$). Similarly, the LII group played less than the LLL group ($t_{205.48} = 2.96$, $p < 0.01$), yet did not differ from the HHH ($t_{205.48} = 1.48$, $p = 0.14$) and III ($t_{205.48} = 1.44$, $p = 0.15$) groups. By contrast, play levels did not differ between heterogeneous groups (HII vs. LII: $t_{205.48} = 0.52$, $p = 0.60$) or between homogeneous groups (III vs. HHH: $t_{205.48} = 0.03$, $p = 0.98$; III vs. LLL: $t_{205.48} = 1.52$, $p = 0.14$; HHH vs. LLL: $t_{205.48} = 1.48$, $p = 0.14$).

## Comparison between homo- and heterogeneous treatment groups

Since the interaction between treatment group and time appeared to be mainly explained by differences between homo- and heterogeneous groups, the development of play levels of these

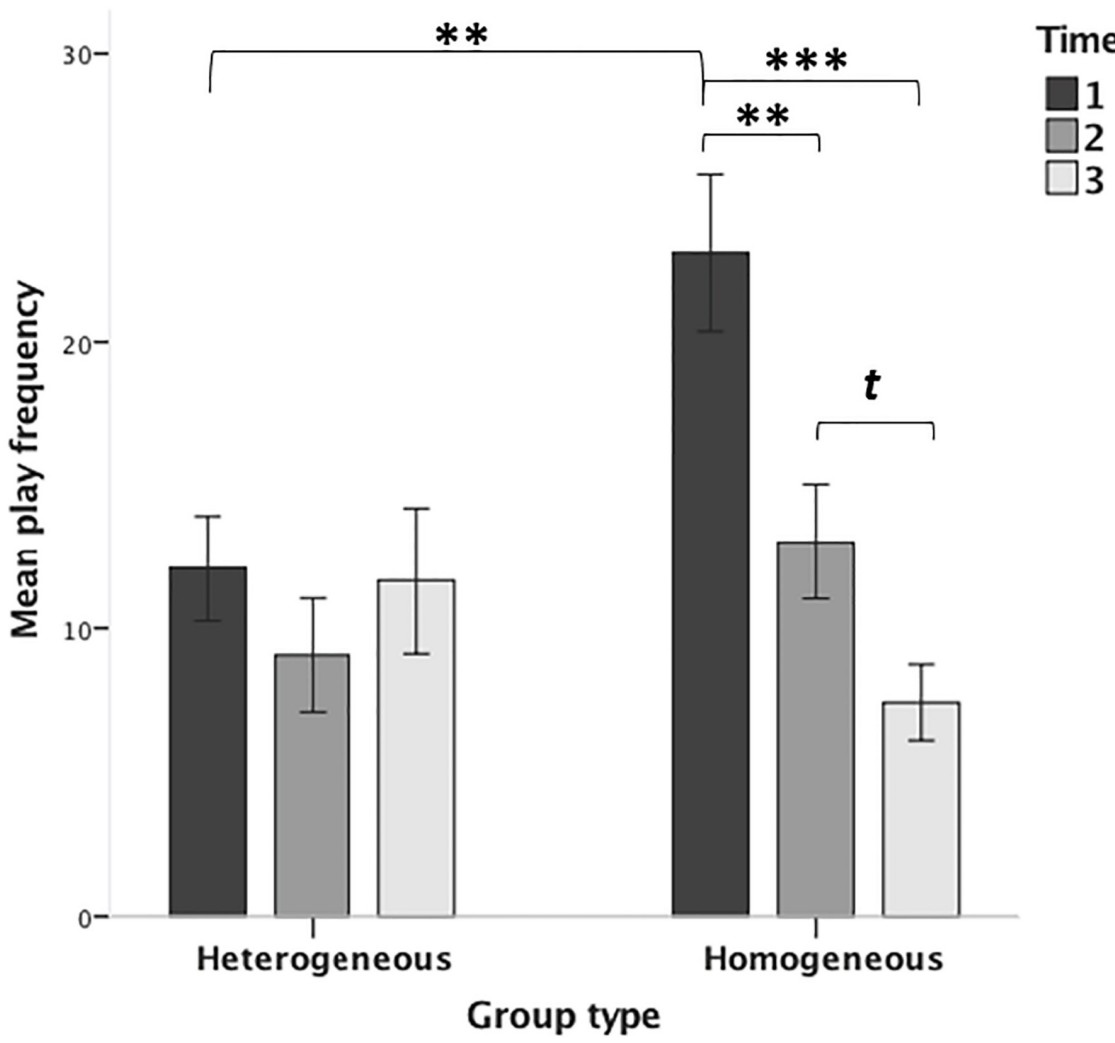

**Fig 4. Histograms showing home cage play levels (pinning frequency) across treatment group types (heterogeneous vs. homogenous) and time; given are means and S.E.M.s.** Brackets indicate differences between time points ($t$: p < 0.1; *: p < 0.05; **: p < 0.01; ***: p < 0.001).

two group types across time was investigated further. Time and group type significantly interacted ($F_{1,150}$ = 14.11, p < 0.001; Fig 4). The main effects of time ($F_{1,150}$ = 15.76, p < 0.001) and group type ($F_{1,204.48}$ = 11.52, p = 0.001) were also significant.

According to the estimates of covariance parameters, there was no significant cage variation (Wald $z$ = 1.42, p = 0.16). Therefore the random cage factor was disregarded in further comparisons.

At Time 1, animals in the homogeneous groups played more than in the heterogeneous ones ($t_{70.92}$ = -3.34, p = 0.001; means ± SD: 23.08 ± 18.30 vs. 12.10 ± 10.10), whereas at Time 2 and 3, the play frequencies of the homo- and heterogeneous groups did not differ (Time 2: $t_{73}$ = -1.66, p = 0.10; Time 3: $t_{49.16}$ = 1.04, p = 0.30). While the play frequency in the homogeneous group type declined across the time points ($F_{2,88}$ = 15.94, p < 0.001; Time 1 vs. 3: p < 0.001; Time 1 vs. 2: p < 0.01; Time 2 vs. 3: p = 0.06), it remained unchanged across time in the heterogeneous group type ($F_{2,58}$ = 1.04, p = 0.36). With regard to the main effect of group type,

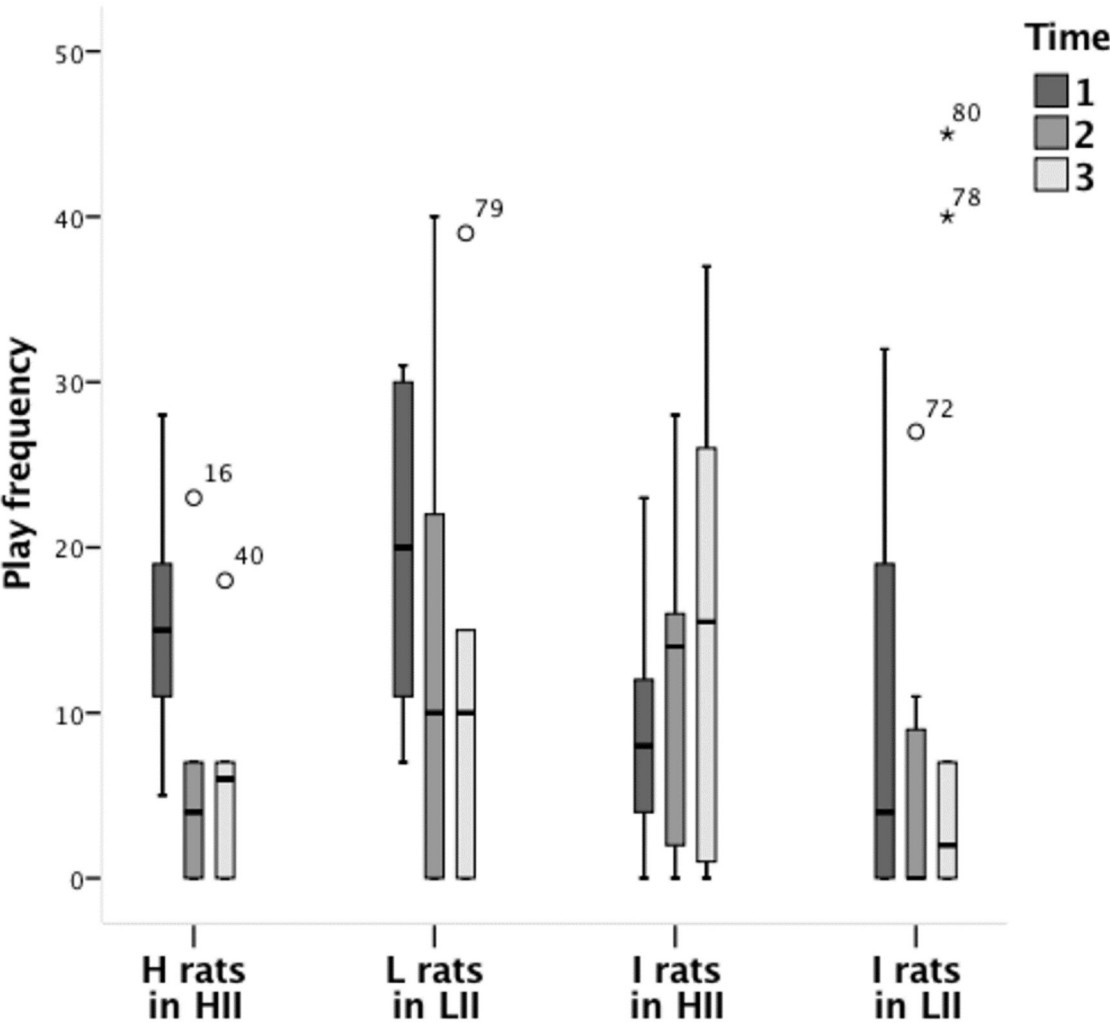

**Fig 5. Boxplots of pinning behaviour frequencies across time for seeders and I rat pairs from the heterogeneous groups.** The box represents the middle 50% of the data, while the upper and lower whiskers show the extreme data points, excluding outliers. The horizontal line within the box represents the median.

the estimates of fixed effects showed that overall rats played more in the homo- than in the heterogeneous groups ($t_{204.48} = 3.40$, p = 0.001).

## Play levels of seeder and intermediate rats within the heterogeneous group type

To better understand the lower overall play levels of the heterogeneous group type, especially at Time 1, the play levels of the seeders were compared with those of the I rat pairs (Fig 5).

The play frequencies of the I rat pairs in the HII group did not differ from those of the I rat pairs in the LII group at Time 1 ($U = 12.0$, $z = -0.10$, p = 1.00), Time 2 ($U = 5.0$, $z = -1.58$, p = 0.14) or Time 3 ($U = 9.5$, $z = -0.62$, p = 0.60). Therefore, these two groups were combined for further comparisons (i.e., 10 seeders vs. 10 pairs of I rats).

The play levels of the seeders decreased across time ($\chi^2_2 = 7.78$, p = 0.02), while those of the I rat pairs did not change over time ($\chi^2_2 = 1.38$, p = 0.52). Furthermore, the play frequencies of seeders and I rat pairs differed at Time 1 ($U = 22.5$, $z = -2.08$, p = 0.04; mean ± SD, seeders:

17.70 ± 9.52; I rat pairs: 9.30 ± 7.22), yet not at Time 2 ($U$ = 50.0, $z$ = 0.00, p = 1.00) or at Time 3 ($U$ = 35.0, $z$ = -1.14, p = 0.26).

## Social preference test

In the heterogeneous groups, there were no significant effects of cage (HII: $\chi^2_4$ = 0.00, p = 1.00; LII: $\chi^2_4$ = 6.66, p = 0.16). One-Sample Wilcoxon Signed Rank Tests indicated that subject rats did not show a preference between the seeder (H or L) and the I stimulus cage mate (HII: p = 0.64; LII: p = 0.72).

## Consistency over time of Play-in-Pairs tests and home cage play

PN frequencies of first and second Play-in-Pairs test were positively correlated ($\underline{r}_{72}$ = 0.23, p = 0.04).

PN frequencies in the home cage at Time 1 were positively correlated with those at Time 2 ($r_{s\,75}$ = 0.32, p = 0.006). However, at Time 1 and Time 3 ($r_{s\,75}$ = 0.12, p = 0.30) as well as Time 2 and Time 3 ($r_{s\,75}$ = 0.01, p = 0.94), they were not correlated (Bonferroni adjustment: $\alpha$ = 0.017). The Spearman test results, however, were affected by a significant cage effect at Time 1 ($\chi^2_{24}$ = 37.28, p = 0.04), with a tendency also at Time 2 ($\chi^2_{24}$ = 35.84, p = 0.06) and Time 3 ($\chi^2_{24}$ = 35.98, p = 0.06).

## Associations between second Play-in-Pairs test and home cage play

There were no significant relationships between PN frequencies in the second Play-in-Pairs test (days 77–79) and PN in the home cage at the closest time points (Time 2: $\underline{r}_{72}$ = -0.10, p = 0.40; Time 3: $r_{s\,75}$ = -0.12, p = 0.29), with cage having no effect on the play frequencies of the second Play-in-Pairs test ($\chi^2_{24}$ = 20.02, p = 0.70).

## Play Role index

The Play Role index was significantly lower in the second Play-in-Pairs test than in the home cage (Play-in-Pairs: 0.25 ± 0.15; home cage: 0.37 ± 0.20; $z$ = -2.49; p = 0.01) indicating greater play reciprocation during Play-in-Pairs. As expected, Play Role index increased over time in the home cage ($\chi^2_2$ = 10.33, p = 0.006; significant Bonferroni-adjusted comparison: Time 1 < Time 3, p = 0.006). However, Play Role index did not change between the first and the second Play-in-Pairs test ($U$ = 296.0, $z$ = 1.39, p = 0.17).

## Discussion

This study investigated how differences in individual playfulness affected social play development across adolescence and early adulthood in rats. Individual playfulness was measured via a Play-in-Pairs test [46, 45], after which rats were resorted into cages of similar (high, HHH; intermediate, III; low, LLL) or dissimilar (HII, LII) playfulness. Our main finding was that, overall, similarity in playfulness among cage mates increased home cage play levels, irrespective of the initial levels of individual playfulness (H, I, or L). More precisely, heterogeneous groups played less than homogeneous groups at adolescence (8 weeks of age), while no significant differences were found during late adolescence / early adulthood (10 and 12 weeks of age). Surprisingly, this result appeared to be mainly driven by the LLL group, which played the most at adolescence. Correspondingly, the play levels of the heterogeneous groups did not differ over time, whereas the homogeneous groups decreased play over time as expected according to the literature. The H and L seeder rats from the heterogeneous groups did not produce the hypothesised differences in social play, and they were not preferred or avoided,

respectively, by the I cage mates in the social preference test. This indicates that emotional contagion between seeder and cage mate rats, if it occurred, did not affect welfare in terms of expressed group play levels; it rather suggests that more complex group dynamics affected play development. Furthermore, a preliminary analysis comparing the play levels of seeder and I cage mates showed that it was the latter which played less at adolescence, further suggesting that the presence of a cage mate with higher or lower playfulness in the group may have been the underlying cause of the main finding in this study.

## Effect of similarity or dissimilarity in playfulness on social play development

Play in rats peaks between 30 and 40 days of age and starts to decline during the 8th to 9th week [56, 8, 44]. As in this study social play in the home cage was measured starting from the 8th week of age, we would have expected to observe a decrease of play over time by all treatment groups. While play levels of the homogeneous groups decreased as expected, play levels of the heterogeneous groups did not differ across time points, mainly because these groups played relatively less shortly after being formed. Such decrease in play was not compensated for in later adolescence / early adulthood. Therefore, similarity in playfulness, rather than individual playfulness levels at resorting of the animals, had a major impact on play development.

That similarity in playfulness promotes relatively higher levels of home cage play concurs with previous personality research in rats and other animal species, including humans. In rats, playful juveniles tend to engage in play more frequently when they have similarly playful and responsive partners [47]. In humans, personality (mis)matches and their repercussions on quality of life have been investigated extensively. Izard [57], for example, found that similarity in personality facilitates the mutual expression of positive affect. Perrone & Sedlacek [58] compared homogeneous and heterogeneous counselling groups and reported a higher level of group cohesiveness and thus a higher level of client satisfaction in the homogeneous groups. The principle of homophily, i.e., similar people are more likely to form affinitive contacts with one another than dissimilar people, represents in fact one of the most pervading factors for determining friendships in humans [59]. Indeed, individuals tend to select friends with similar levels of personality dimensions such as agreeableness, extraversion and openness [60]. In chimpanzees, friends or individuals who sat together more often showed more similarity in sociability and boldness than non-friends [61]. The authors of that study suggested that the preference for similar personalities may contribute to a decrease of uncertainty within interactions, giving a sense of reliability, which would be especially important in cooperative contexts. Also, a recent study in horses demonstrated that unrelated individuals assort themselves according to personality [62]. Similar results have been found also across species, since dogs and their owners showed similarities in the personality dimensions of neuroticism, extraversion, conscientiousness, agreeableness, and openness [63]. Thus, a higher level of compatibility among more similar partners seems to facilitate positive experiences and to promote cooperative behaviour. Our findings suggest that also for playfulness, compatibility between individuals promotes positively valenced experiences via the increase of overall social play.

In the attempt to at least partly unravel the underlying mechanisms through which more heterogeneous groups played less during the first time period after resorting, the play frequencies of seeders and intermediate rats in the heterogeneous groups were compared. Play frequencies of the seeders decreased across time, and were greater than those of the intermediate rats at Time 1, yet did not differ from them at Times 2 and 3. Correspondingly, play frequencies of the intermediate rats did not differ across time. Therefore, the intermediate cage mates of the seeder rats were the animals more negatively affected by their group composition,

indicating that it was the presence of a more (H) or less (L) playful cage mate that inhibited overall play levels in heterogeneous groups. Adolescent rats that were isolate-housed and thus highly motivated to play engaged in high levels of play when paired with group-housed rats of the same age, however the group-housed partners also showed more avoidance of the perhaps "too play—motivated" isolate-housed rat [9, 44]. While this evidence may explain our finding that play frequencies of intermediate rats were initially inhibited by overly motivated, playful seeders (HII group), it does not explain why the same outcome occurred also when the seeders were low playful rats (LII group). Isolate-housed rats paired with less socially active group-housed partners reduced play behaviour and increased other social behaviours unrelated to play [44]. Similarly in our study, it is possible that a low playful seeder was not perceived by its cage mates as an interesting partner to play with, at least during the initial time period after resorting. As play levels between seeder and intermediate rats did not differ in later adolescence / early adulthood, it could more simply be that more dissimilar / incompatible partners require more time to adjust to each other.

The unexpected finding that rats from the LLL group not only played as much as rats from III and HHH groups across adolescence, but also (from visual inspection of the data) appeared to be the animals that played the most, especially in the first time period after resorting, highlights the important effects that social manipulations can have on personality development. It has been shown that rats displaying low play levels with control, saline-treated play partners tended to engage in higher levels of play initiation when paired with non-playful, scopolamine-treated rats [7]. Similarly, it appears that the drive to play of L rats may have been inhibited by dissimilar / more playful cage mates in the time period before resorting, and instead could be better expressed with cage mates having a more similar playfulness level after resorting. However, the fact that the playfulness scores in our study derived from only three play encounters between the subject rat and its three cage mates attenuates, but does not exclude the possibility that such scores were influenced by the specific playfulness of the cage mates. One way to overcome this problem would be to refine the Play-in-Pairs test (e.g., by increasing the number of random, coetaneous partners used in the play encounters) to further improve the assessment of an individual´s playfulness.

Further research should focus not only on changes in play quantity, but also in play quality, derived by resorting of rats based on playfulness. For example, rats with certain types of brain damage initiate just as many attacks and are as likely to defend themselves as their intact controls, but the intact rats tend to avoid initiating play with such rats. This indicates that the brain-damaged rats are not playing in a manner found to be appropriate by the partner and thus they become less attractive play mates [64]. Similarly, discordance in how the players may have interacted in the heterogeneous groups could be at the basis of the observed differences in play amount between homogenous and heterogeneous groups. In relation to play quality, it should be noted that in this study only the attack component of social play (i.e. the act of pinning and the attack to the nape) was used as a proxy measure of playfulness, thus assuming that more playful rats are also more likely to respond to such an attack (i.e., defend) and use tactics of defence that facilitate prolonged playful contact [8]. This may not always be the case as shown by previous research [65, 66]. However, frequencies of pinning and of being pinned (i.e., performing a complete rotation as a defence strategy) were found to be positively correlated in Lister Hooded rats [45], suggesting that the attack component of play can approximate the defence component well in this strain. Other aspects to be considered are that AN, as defined in this study, can underestimate the number of attacks launched, since rats can begin to defend against a nape attack before contact with the nape is made [8]. Also, the adoption by the play partner of defensive strategies other than complete rotation, such as partial rotation and evasion, may affect the quality of a play encounter. However, the initiations of social play

in adolescent Lister hooded rats were shown to almost exclusively lead to pinning behaviour, with very few instances of other playful defensive strategies [46, 45]. Taken together, the main outcome measures in our study seem to represent playfulness personality well, yet it has to be taken into account that in other strains both AN and PN may not be tightly correlated with overall attack and defence behaviours [65].

### Emotional contagion between seeders and their cage mates

If emotional contagion between the seeders and their cage mates had affected group play development at any point during adolescence / early adulthood, the group with the highly playful seeder (HII) would have played more than the group with the low playful seeder (LII), however this was not the case. The absence of a long-lasting effect of emotional contagion on group play was supported by the results from the social preference test, which showed that intermediate rats did not prefer to spend more time with highly playful cage mates, and did not avoid less playful cage mates. However, the failure to reveal a significant preference may be more a reflection of the test paradigm used. At this age, rats may be similarly likely to play with both familiar and unfamiliar partners and so, when confronted with rats to remain in close proximity, the cues may not be sufficient for the rat to discriminate. A conditioned place preference test, in which the I rat has a choice between a chamber that involves prior play with an H partner or L partner versus a chamber that involves prior play with an I partner, may be a more powerful tool to determine whether I rats have a preference.

While in the heterogeneous groups the seeders initially played more than the intermediate rats, in later adolescence play levels no longer differed between the two categories of rats. Since personality, especially during adolescence, is susceptible to social influences and may be regarded as only temporarily stable [67], it is possible that by the time the social preference test occurred (week 16 of age), the individual playfulness of the rats had changed. This could be an alternative explanation of why social preferences were not detected in this test. However, play levels in the Play-in-Pairs tests showed a weak yet significant consistency across adolescence (weeks 5 and 12 of age) indicating that to a certain extent and despite a strong manipulation of the social environment in between the two tests, underlying individual differences in playfulness were partly maintained. One previous study in which a similar Play-in-Pairs test was performed, yet without the social disruption of resorting, showed even higher consistency of play in pairs across time (for PN, [45]: $r_{21}$ = 0.50, p = 0.02; the present study: $r_{72}$ = 0.23, p = 0.04).

### Consistency of home cage play and relationship with play in pairs

Home cage play levels were positively related between Time 1 and Time 2. However, variation between cages was large and there were no relationships between the remaining time point combinations. Therefore, we conclude that the home cage observation time intervals may not have been sufficient to detect playfulness differences at the individual level, contrary to the Play-in-Pairs tests.

Individual playfulness as measured by the Play-in-Pairs test and home cage play levels were not related. This may have been determined by the resorting procedure, which may have partially affected the personality development of the rats. There are contrasting findings in the literature on the lack [45] or presence [46] of a relationship between play in pairs and home cage play, likely due to specific housing conditions of each experiment. This study confirms that the Play-in-Pairs test is a better predictor of playfulness compared to more time-consuming home cage observations.

## Play Role index across age and play contexts

Play of male rats does not only decrease in quantity with age but also changes in quality, mainly due to a shift from adopting defensive strategies that promote play (e.g., complete rotation into supine position) in the juvenile / adolescent period to performing defensive behaviours that discourage the continuation of a play bout (e.g., evasion or partial rotation of the body) during adulthood [8, 68]. As hierarchy formation is likely to be at the basis of such changes, it is possible that playful interactions performed by younger rats may be experienced differently compared to those performed by adults, potentially questioning the positive valence of play in early adulthood. The Play Role index used in our study provided an indication of play symmetry between play partners, i.e. the extent to which playful pinning was reciprocated between them. Considering that an index of 0 indicates full play reciprocation and an index of 1 indicates that play is unidirectional, both home cage play and, in particular, play in pairs following a short social isolation showed relatively high play symmetry (indexes of 0.37 and 0.25, respectively), more similar to what would be expected during adolescence than in adulthood. Surprisingly, the Play Role index for play in pairs did not change between the juvenile and early adulthood, suggesting that this test may induce adults to play more like juveniles. Instead, home cage play became increasingly asymmetrical with age thus it remains to be investigated whether such change in play quality may reflect a less "enjoyable" experience by at least some animals. However, our main finding that homogenous groups played more than heterogeneous ones refers to the adolescence period, when play in the home cage still showed a relatively high level of play reciprocity, suggestive of a play environment still largely unaffected by hierarchy formation.

## Conclusions and implications for animal welfare

Our study showed that similarity in playfulness, rather than long-term emotional contagion processes, increased home cage play levels. Such increase occurred for every level of playfulness as long as group mates were similarly playful, highlighting the importance of social group composition and compatibility between individuals for the outcome of experiments and for animal welfare. As play, especially during adolescence, has been proposed as a proxy measure of positive affect [13, 4, 14], our results suggest that a mismatch in playfulness may negatively affect the development of group play and, in turn, animal welfare.

## Supporting information

**S1 Appendix. ARRIVE guidelines checklist.**
(PDF)

**S1 Dataset. Dataset.**
(XLSX)

**S1 Table. Description of the main phases of the experiment.**
(DOCX)

**S2 Table. Classification of rats into high (H; n = 20), intermediate (I; n = 35) and low (L; n = 20) playfulness categories, and assignment to treatment groups and cages.**
(DOCX)

## Acknowledgments

We would like to thank Dr. Bernhard Voelkl and Dr. Jeremy Bailoo for statistical advice, and Kathryn Finlayson for participation in animal care and handling. Our thanks also go to Zeljko Kragic for technical assistance with the experimental setup.

## Author Contributions

**Conceptualization:** Jessica Frances Lampe, Oliver Burman, Hanno Würbel, Luca Melotti.

**Data curation:** Jessica Frances Lampe, Luca Melotti.

**Formal analysis:** Jessica Frances Lampe, Sabrina Ruchti, Luca Melotti.

**Funding acquisition:** Hanno Würbel, Luca Melotti.

**Investigation:** Jessica Frances Lampe, Sabrina Ruchti, Luca Melotti.

**Methodology:** Jessica Frances Lampe, Oliver Burman, Hanno Würbel, Luca Melotti.

**Project administration:** Jessica Frances Lampe, Luca Melotti.

**Supervision:** Jessica Frances Lampe, Hanno Würbel, Luca Melotti.

**Visualization:** Jessica Frances Lampe, Sabrina Ruchti.

**Writing – original draft:** Jessica Frances Lampe, Luca Melotti.

**Writing – review & editing:** Jessica Frances Lampe, Sabrina Ruchti, Oliver Burman, Hanno Würbel, Luca Melotti.

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
