## [Decision Letter · Decision Letter 0]

4 Sep 2019

[EXSCINDED]

PONE-D-19-21615

Play like me: similarity in playfulness promotes social play

PLOS ONE

Dear Dr Melotti,

Thank you for submitting your manuscript to PLOS ONE. After careful consideration, we feel that it has merit but does not fully meet PLOS ONE’s publication criteria as it currently stands. Therefore, we invite you to submit a revised version of the manuscript that addresses the points raised during the review process.

Both the reviewers agree that the paper is much improved. However, both of them ask for additional clarifications about definitions of some theoretical concepts. 

I agree with them, for this reason I suggest the authors to do the changes requested which are really minor, so I do not think that it will be a problem to address them.

We would appreciate receiving your revised manuscript by October 30th, 2019. To enhance the reproducibility of your results, we recommend that if applicable you deposit your laboratory protocols in protocols.io, where a protocol can be assigned its own identifier (DOI) such that it can be cited independently in the future. For instructions see: http://journals.plos.org/plosone/s/submission-guidelines#loc-laboratory-protocols

We look forward to receiving your revised manuscript.

Kind regards,

Elisabetta Palagi

Academic Editor

PLOS ONE

2. Please complete and submit a copy of the ARRIVE Guidelines checklist, a document that aims to improve experimental reporting and reproducibility of animal studies for purposes of post-publication data analysis and reproducibility: https://www.nc3rs.org.uk/arrive-guidelines.

Please include your completed checklist as a Supporting Information file. Note that if your paper is accepted for publication, this checklist will be published as part of your article.

Reviewers' comments:

Reviewer's Responses to Questions

**Comments to the Author**

1. Is the manuscript technically sound, and do the data support the conclusions?

Reviewer #1: Yes

Reviewer #2: Yes

2. Has the statistical analysis been performed appropriately and rigorously? 

Reviewer #1: Yes

Reviewer #2: Yes

3. Have the authors made all data underlying the findings in their manuscript fully available?

Reviewer #1: Yes

Reviewer #2: Yes

4. Is the manuscript presented in an intelligible fashion and written in standard English?

Reviewer #1: Yes

Reviewer #2: Yes

5. Review Comments to the Author

Reviewer #1: The present version of the paper is much improved with the core findings now being more readily accessible. The core finding of importance is that heterogeneous groups of rats play less than do homogeneous ones. The unexpected finding was a failure to produce a consistent contagion effect. These major findings are now thoroughly discussed and their importance made clear. Less convincing is the ability to select for consistently high playing animals and how best to characterize behavioral differences across individuals and groups. I highlight these methodological issues below. In part, the problem is well beyond the confines of the present paper. Social play is, by its very nature, a social behavior, so isolating the contribution of one individual from that of the dynamics created by interacting with another animal is extremely hard. The approach used in the present paper is only partially successful, but the limitations that emerge are in themselves instructive. Hopefully, this paper will inspire others to explore different ways of identifying individuals with a different propensity to play in a way that can be dissociated from the influence of the partner. For example, repeatedly testing a subject with a standardized partner, such would be produced by injecting the partner with scopolamine. I have no further substantive concerns, but I do note some items that could be considered further in the Discussion to increase the impact that this paper. Certainly, some of these concerns should be considered in future studies.

Lines 145-146: Not clear why the ‘one or more’ is important for the prediction that “…(iv) one or more homogenous groups would play more than the heterogeneous groups…” Why not homogeneous groups should play more than heterogeneous groups? Clarify.

Lines 183-186: The limitation of the method used is that, as the original grouping of rats is random, the multiple pair tests can only assess the relative playfulness of the individuals chosen to play together. As noted in the Introduction, there is evidence that the playfulness of the partner can increase or decrease the level of play expressed by an individual in a pair. While this procedure can attenuate the influence of the partners in assessing an individual’s playfulness (as described in lines 203-210), it does not eliminate that influence. Therefore, the labeling of rats as high or low players is relative, so that at a population level, it is not necessarily the case that that an H rat derived from a pair test would be on the extreme of the distribution representing H players. That these designations may not be robust is shown by the subsequent findings. The highest play frequency recorded is for the LLL group, especially on day 1 (Figure 3). Taking into account the variance, the statistical analyses show that there is no difference between the homogeneous groups (lines 414-416). This is confusing, because if the groups were formed by selecting the highest players and the lowest players, then how come the groups made up of high players are not playing more than the group constituted of low players? If anything, the low playing group is tending to play more (Figure 3). In this context, it is hard to understand what it means to be selected as a high or low player from the pair tests. As the consistency in a playful personality is central to the present paper, these discrepancies need to be adequately addressed. While these unexpected findings are discussed in terms of the effects of social manipulations on personality development (lines 573-577), they also need to be considered with regard to methodology. Perhaps the approach used to select more and less playful rats is not as robust as hoped. Although the lack of consistency between the pair-test and home cage levels of play is noted (lines 626-633), the broader issue of how best to identify high and low players selectively in a robust manner has not been resolved.

Lines 229-233: As defined, AN can underestimate the number of attacks launched, and PN can confound the role of the attacker with that of the attacker. Rats can begin to defend against a nape attack before contact with the nape is made, and by the definition given, these would not be counted as AN. Most pins arise when the recipient of an attack rotates to supine to protect its nape from being contacted (see Pellis & Pellis, 1987 cited), therefore pinning is not a property of the attacker alone, but the combination of its attack with the defensive maneuver used by the defender. For the latter, the use of the “Play Role Index” as a measure of relative asymmetry (lines 234-238) partially overcomes this problem as the rotated state of both pair mates is taken into account. Similarly, based on our recent experience with Lister hooded rats, it appears that as for Long Evans hooded rats, the correlation between attack and the specific attack outcome as defined by AN is quite tight, as is the correlation between overall defense and rotating to supine. However, in other strains, both AN and PN are not so tightly correlated with attack and defense (see Himmler et al., 2016 cited). For laboratories using other strains of laboratory rats, the methods used in the present paper with Lister hooded rats may not be as capable of detecting the effects of individual differences in playfulness. It would be worth noting these limitations in the Discussion to avoid the generalizability of the present findings from being compromised.

Lines 379-381: It is to be expected that AN and PN are correlated, as the supine position is most often achieved when the recipient of a nape attack rotates to supine to protect its nape from being contacted. However, the finding that PN is more frequent than AN is misleading. It is extremely rare that a rat rotates to supine in the absence of its partner launching an attack. Therefore, the discrepancy must arise because of the restricted measure of attack as defined by AN. As noted above, defense can be initiated before the attacker makes contact (see Himmler, B. T., Stryjek, R., Modlińska, K., Derksen, S. M., Pisula, W., & Pellis, S. M. (2013). How domestication modulates play behavior: A comparative analysis between wild rats and a laboratory strain of Rattus norvegicus. Journal of Comparative Psychology, 127, 453-464; Himmler, S. M., Modlińska, K., Stryjek, R., Himmler, B. T., Pisula, W., & Pellis, S. M. (2014). Domestication and diversification: A comparative analysis of the play fighting of the Brown Norway, Sprague-Dawley, and Wistar strains of laboratory rats. Journal of Comparative Psychology, 128, 318-327), which can lead to a rat beginning to rotate to supine before the AN configuration is achieved. Again, depending on the strain of rat used, this discrepancy can make a difference in how accurate either AN or PN is as a measure of attack (see Himmler et al., 2016 cited).

Lines 585-593: The issue of play quality may also be important in interpreting the main effect shown in this study – that heterogeneous groups play less than homogeneous groups. For example, rats with certain types of brain damage initiate just as many attacks and are as likely to defend themselves as their intact controls, but the intact rats tend to avoid initiating play with such rats. Clearly, the brain-damaged rats are not playing in a manner found to be appropriate by the partner and so they become less attractive play mates (Pellis, S. M., Hastings, E., Shimizu, T., Kamitakahara, H., Komorowska, J., Forgie M. L., & Kolb, B. (2006). The effects of orbital frontal cortex damage on the modulation of defensive responses by rats in playful and non-playful social contexts. Behavioral Neuroscience, 120, 72-84). What such findings show is that compatibility in the amount of play initiated is only one factor of importance, the other is how the partner plays. While the present study makes a convincing case that the amount of play initiated by different rats is being reasonably well measured, it does not capture possible differences in the style of play. For the heterogeneous groups, discordance in how the players interact may be more important than in differences in how much play they initiate. This may be a useful avenue to explore in future studies.

Line 725: “J Comp Psychol” should be “Int J Comp Psychol”

Reviewer #2: I think that the authors have made a good work with the revision of the MS by following all my suggestions. However, I still have concerns about the definition of some theoretical concepts. Before completely accepting the MS, I would like the authors address the following doubts.

Line 69: not only continuous interaction but also (and mainly) tight physical contact

Lines 71-77: there is still confusion between the concepts of emotional contagion, Rapid Mimicry (RM) and behavioural synchronization. Emotional contagion occurs when a subject not merely perceives but mainly shares the same affective state of another individual. Certain forms of emotional contagion, but not all (e.g. consolation), can be mediated by mimicry, one of the phenomena at the basis of the behavioural synchronization. Since the mimic response is extremely rapid (< 1 sec), it occurs outside the conscious awareness and voluntary control. RM involve the Mirror Neuron System (MNS). Basing on the Perception-Action Model (Preston & de Waal, 2002), the involvement of MNS during the perception of an action or a facial expression activates shared representations: through the RM of an observed behaviour, emotional state related to that behaviour may be arisen in the observer.

Lines 78-83: the relation between emotional contagion and animal welfare is not still completely clear. Some studies have evidenced that the frequency of RM increases with the level of familiarity between subjects. For example, humans are likely to mimic more frequently the facial expressions of in-group- than out-group members (Seyfarth & Cheney, 2013). Moreover, within group RM is affected by the relationship quality of the interacting subjects (in humans, friends mimic each other’s smiles more than non-friends, Fisher et al., 2012). In this light, the replication and the sharing of fellows ' emotions promotes the development of tighter social relationships that contributes to social cohesion, one of the factors promoting animal welfare.

Lines 508-509: I agree with the statement that different factors can affect play development, but the absence of evidence is not the evidence of absence. The fact that the presence of H and L seeders in the heterogeneous groups did not affect the play levels of I cage mates does not indicate the absence of emotional contagion during playful interactions. The authors investigated only PN and AN but they did not take into account other play patterns and play signals that can suggest the occurrence of the emotional contagion. I think that there is a still confusion between the concept of emotional contagion and playfulness personality.

6. PLOS authors have the option to publish the peer review history of their article (what does this mean?). If published, this will include your full peer review and any attached files.

Reviewer #1: No

Reviewer #2: No

---

## [Author Response · Author response to Decision Letter 0]

8 Oct 2019

Responses to the Reviewers´ comments are provided in the uploaded Response to Reviewers file.

---

## [Editor Report · Decision Letter 1]

10 Oct 2019

Play like me: similarity in playfulness promotes social play

PONE-D-19-21615R1

Dear Dr. Melotti

We are pleased to inform you that your manuscript has been judged scientifically suitable for publication and will be formally accepted for publication once it complies with all outstanding technical requirements.

With kind regards,

Elisabetta Palagi

Academic Editor

PLOS ONE

---

## [Editor Report · Acceptance letter]

17 Oct 2019

PONE-D-19-21615R1 

Play like me: similarity in playfulness promotes social play 

Dear Dr. Melotti:

I am pleased to inform you that your manuscript has been deemed suitable for publication in PLOS ONE. Congratulations! Your manuscript is now with our production department. 

With kind regards,

on behalf of

Dr. Elisabetta Palagi 

Academic Editor

PLOS ONE